

# Russia's black carbon emissions: focus on diesel sources

**N. Kholod[1], M. Evans[1], T. Kuklinski[2]**

[1]Joint Global Change Research Institute, Pacific Northwest National Laboratory, 5825 University Research Court, Suite 3500, College Park, MD 20740, USA

[2]US Environmental Protection Agency, Office of International and Tribal Affairs, 1200 Pennsylvania Ave., NW, Washington, DC 20460, USA

*Correspondence to:* N. Kholod (Nazar.Kholod@pnnl.gov)

**Abstract.** Black carbon (BC) is a significant climate forcer with a particularly pronounced forcing effect in polar regions, like the Russian Arctic. Diesel combustion is a major global source of BC emissions, accounting for 25–30 % of all BC emissions. The demand for diesel is growing in Russia, but Russian diesel emissions are poorly understood. This paper presents a detailed inventory of Russian BC emissions from diesel sources. Drawing on a complete Russian vehicle registry with detailed information about vehicle types and emission standards, this paper analyzes BC emissions from diesel on-road vehicles. We use the COPERT emission model with Russia-specific emission factors for all types of on-road vehicles. On-road diesel vehicles emitted 21 Gg of BC in 2014; heavy-duty trucks account for 70 % of the on-road BC emissions, while cars represent only 4 % (light commercial vehicles and buses account for the remainder). Using Russian activity data and fuel-based emission factors, the paper also presents BC emissions from diesel locomotives and ships, off-road engines in industry, construction and agriculture, and from diesel generators. The study also factors in the role of superemitters in BC emissions from diesel on-road vehicles and off-road sources. The total emissions from diesel sources in Russia are estimated to be 48 Gg of BC and 16 Gg of OC in 2014. Off-road diesel sources emitted 57 % of all diesel BC in Russia.

## 1    Introduction

Black carbon (BC), a component of particulate matter (PM), may be the second most powerful contributor to climate change after carbon dioxide (Bond and Sun, 2005; Bond et al., 2013). BC emissions cause significant warming effects through direct absorption, interaction with clouds, and reduced albedo on snow. BC plays an important role in polar regions because when it deposits on snow and ice, it facilitates the absorption of solar radiation, increases air temperature, and accelerates snow and ice melting (Warren and Wiscombe, 1980; Hansen, 2003;Quinn, 2008; Quinn et al., 2011; Koch and Del Genio, 2010; Bond et al., 2013).

BC, as a major component of diesel PM, also has adverse health impacts (WHO, 2012). Chronic exposure to PM is associated with a range of diseases, as well as premature death from cardiopulmonary disease and lung cancer (Pope III et al., 2011). The World Health Organization estimates that PM pollutions accounted for 3.1 million deaths in 2010 (WHO, 2013). Exposure to PM emissions is the ninth leading factor of premature death globally (Lim and et.al., 2012).



BC is a product of incomplete combustion of fossil fuels, biofuels and biomass. This study focuses on BC emissions from diesel sources only per the parameters of the United States Environmental Protection Agency's (US EPA) Reduction of Black Carbon from Diesel Sources in the Russian Arctic project, part of the USG Arctic Black Carbon Initiative (EPA, 2016). Diesel BC emissions accounted for 25-30 % of all global energy-related BC emissions in 2000 (Bond et al., 2013). On-road and off-road diesel transport accounts for about 70 % of all energy-related BC emissions in Europe, North America, and Latin America. Diesel is important to BC for two additional reasons. First, diesel combustion results in a high share of BC compared to organic carbon (OC). Since OC has a net cooling effect, the ratio of BC to OC is critical to the forcing effect of emissions. Second, there are well-established control technologies and policies to reduce emissions from diesel combustion. In fact, this study finds Russian emissions are lower than many previous studies specifically because we are able to account for the impact of the emission standards in reducing emissions.

Russia is the second largest producer of crude oil in the world (IEA, 2015a), likewise it is a large producer and consumer of diesel fuel. Diesel production has risen over the last decade by 10 % (Minenergo, 2015). Despite the large diesel consumption, Russia has historically represented a significant gap in our global understanding of BC emissions.

The aim of this paper is to present a detailed BC emission inventory from diesel sources in the Russian Federation for the base year 2014. Global or regional emission inventories tend to rely on simple assumptions for emission calculations because country-level data might not be available.

Inventories are also important to developing mitigation policies and improving our understanding of global climate in models. Previous studies have estimated BC emissions from Russian diesel sources primarily by combining fuel consumption with fuel-based emission factors. This paper draws on detailed data about the Russian vehicle fleet, annual mileage by vehicle category, the usage of emission controls on off-road vehicles, and other factors. It also accounts for the existence of super-emitters in the on-road diesel fleet.

## 2    Methodology

### 2.1    Emission standards for diesel engines

Russia has adopted the vehicle emission standards developed by the United Nations Economic Commission for Europe (UNECE). Specifically, Russia introduced Euro 2 standards in 2006 (Supplement S1). Emission standards in Russia apply to both domestically produced and imported vehicles. The minimum emission standard in 2014 was Euro 4 for cars and Euro IV for trucks and buses. As of January 2016, the minimum standard is Euro 5/Euro V. Russia has no official plans to move to the Euro 6 standard.

Russia has also adopted the European standard for PM emissions for off-road vehicles used for agricultural and forestry (these off-road standards are broken into "Stages" in the European system). According to Technical Regulation 031/2012, agricultural and forestry off-road vehicles should meet the UNECE standard (Stage III) (Eurasian Commission, 2012). However, in the baseline year for this inventory (2014), there was no PM emission standard for agricultural vehicles in Russia. It is also appears the implementation of this new standard may be weak.



Except agriculture and forestry, equipment in other sectors in Russia is not obliged to comply with any PM emission standards (Eurasian Commission, 2011a).

## 2.2 Methodology for emission calculations from on-road vehicles

The European COPERT model (COmputer Programme to calculate Emissions from Road Transport) calculates emissions from on-road vehicles. The European Environmental Agency (EEA) supported the development of COPERT. EEA member countries use this free software for official road transport emission inventory preparation (Emisia, 2015). Twenty two countries of the European Union use the model to calculate road transport emissions (Ntziachristos et al., 2009). In this study, we use the COPERT 4 model (version 11.3).

Diesel vehicles in the COPERT model are classified into 4 large groups: passenger cars, light commercial vehicles (LCVs), heavy-duty trucks, and buses. In the model, exhaust BC emissions from vehicles depends primarily on: 1) the number of vehicles on the roads, 2) emission factors, and 3) average number of kilometers traveled.

The COPERT model uses European-specific emission factors and allows users to input country-specific emissions factors. The Russian Scientific Research Institute for Automobile and Transportation (NIIAT) developed

two methodologies for calculating emissions from on-road transport (NIIAT, 2006, 2012). They are based on the simplified EMEP/CORINAIR approach (now called EMEP/EEA Air Pollutant Emission Inventory Guidebook).

NIIAT has developed PM emission factors for Russian vehicles based on vehicle emission test data for both Russian and foreign vehicles. Supplement Table S2 provides the emission factors for the following categories of diesel vehicles: cars, light trucks and buses (LCVs), heavy-duty trucks, and buses on three types of roads (urban,

rural and highways).

We derive BC/PM ratios from the EEA emission guidebook (EEA, 2013). The COPERT model can also directly calculate emissions of elemental carbon (EC) and organic carbon (OC). In European emission studies EC is assumed to be equal to BC for transport due to the nature of the combustion processes.

Diesel engines emit BC emissions in two distinct stages of operations – cold start and hot operation. The

COPERT and the NIIAT models use different approaches to calculate cold start emissions. However, in their study of black carbon emissions from diesel vehicles in the Murmansk Region, Evans et al. showed that both methodologies yield very similar total emissions (Evans et al., 2015). As a result, in this study we do not analyze hot and cold emissions separately and present only the total emissions.

## 2.3 Methodology for emission calculations from off-road engines

BC emission calculations from off-road diesel engines can be expressed by Eq. 1:

$$BC\ emissions = fuel(kg) * PM\ emission\ factor\ (g\ kg^{-1}) * BC/PM\ ratio \qquad (1)$$

We used Formula 1 to calculate emissions from off-road sources listed in Section 5. Table 1 presents PM emission factors and BC/PM speciation ratios used in this study.




Though emission factors reported in the literature could be different from those in the emission guidebook, we use the EEA emission factors for consistency. The advantage of using this approach is that the guidebook reports emission factors for equipment without emission controls. Since the majority of diesel off-road vehicles and equipment is old and Russia does not regulate particulates from off-road diesel sources, we assumed that there are

no emission controls on most of the off-road diesel sources. Some small percentage of imported engines might have emission controls.

**3        Diesel consumption**

**3.1        Production of diesel fuel**

The Russian Federation is the second largest producer of crude oil in the world, producing 13 % of total world oil in

2014 (IEA, 2015a). It is also a large producer of diesel fuel as well. The country increased diesel production from 70 million metric t (Mt) in 2010 to 77 Mt in 2014. Importantly, over the same time frame, Russia implemented standards that improved the quality of its diesel fuel, reducing sulfur content (**Error! Reference source not found.**).

The share of Euro 5 diesel (with sulfur content of 10 ppm, otherwise known as ultralow sulfur fuel) increased from 6 % in 2011 to 50 % in 2014 (Fedstat, 2015g). By the end of 2015, according to an estimate of the Russian

Ministry of Energy, Euro 5 diesel accounted for 82 % of the total diesel production (Government of Russian Federation, 2015). Russia exports more than half of diesel produced; the quality of exported diesel is lower than that of the diesel for the domestic use.

Russia banned the domestic sale of lower grade diesel in 2013. In 2014, only Euro 4 (50 ppm) and Euro 5 (10 ppm) fuel was legal for sale for on-road transport (Eurasian Commission, 2011b). However, compliance with this

standard is not universal. Rosstandard, a government agency responsible for fuel quality control, found that the sulfur content exceeded the maximum allowable content in 21 % of the fueling stations checked in 2014 (Rosstandard, 2015). Though Russia banned high-sulfur diesel, the demand for cheap diesel remains because older engines, especially off-road vehicles, can save money by using high-sulfur diesel.

**3.2        Diesel consumption**

According to the Russian Ministry of Energy, Russia's domestic diesel supply was 32 Mt in 2014 (Minenergo, 2015). Euro 4 and Euro 5 diesel accounted for 88 % (28 Mt) of the domestic diesel supply in 2014, which is more than enough to fuel all the Euro 4 and Euro 5 vehicles. In other words, fuel quality alone likely does not impact emissions.

On-road transport is the largest consumer of diesel, but estimates vary. There are several data sources on diesel

consumption by on-road transport, including official statistics, officially commissioned fuel balances, data from international organizations, and bottom-up estimates.

The Russian fuel consumption statistics are based to a large extent on reports from enterprises. Only medium and large enterprises must report their fuel use to the Federal Statistics Service (Fedstat). Large companies must complete so called TER 4 form on fuel consumption "Fuels and energy inventory, inflow, consumption, and balance



at the end of the reporting period." The aggregated data are publicly available (Fedstat, 2015f, e). Another data source is TER 11 form on fuel consumption by unit of production; however, this information is not available for all sectors. Small businesses are not required to submit this information, yet they employ 11 million people (Fedstat, 2015a, c, b, d). There is no official information on diesel consumption by vehicles owned by individuals either (for example, on diesel sold at fueling stations). As a result, the official data on diesel consumption by on-road vehicles

are incomplete. However, large enterprises do not need to conduct surveys to analyze their sales, so the diesel data likely underreport diesel consumption by individuals and small enterprises.

According to Fedstat, vehicles owned by mid-size and large enterprises consumed 12.7 Mt of diesel in 2013 and 12.2 Mt in 2014 (Fedstat, 2015e). The International Energy Agency (IEA) reports that Russia's on-road transport consumed 11.2 Mt of diesel in 2013 (IEA, 2015b). Both assessments likely underestimate the diesel consumption by

on-road vehicles in the country.

There are several independent bottom-up estimates of diesel consumption by on-road transport. The Russian research company Petromarket estimated that on-road vehicles consumed 23.5 Mt of diesel in 2013 and 24.5 Mt in 2014. Analytical agency Avtostat calculated that on-road vehicles consumed 25.8 Mt of diesel in 2014 (Avtostat, 2015a). Avtostat also estimated that vehicles in the European part of the Russian Federation consumed 70 % of fuel

used by on-road transport. However, their assessments both have their limitations because they do not use a fuel balance approach.

In this paper, we use data from the Russian Center for Energy Efficiency (CENEf) which uses a balance approach for assessing the fuel consumption. CENEf is a leading Russian energy research organization commissioned by the Ministry of Energy to develop fuel balances. It has access to multiple types of fuel statistics

and also uses a sophisticated transport model to calculate fuel consumption by on-road vehicles. CENEf prepares the Russian energy balances by integrating the reporting forms from medium and large enterprises and bottom up calculations to cross-check allocations across sectors. CENEf also makes sure that supply of crude oil and oil products balances demand. Table 2 shows the diesel consumption in 2014 (in thousand metric tons (kt).

CENEf estimated that the total diesel consumption in the country was about 32 Mt in 2014. On-road vehicles

used 22.2 Mt (73 %) of the final diesel consumption. Other significant consumers of diesel are rail, diesel generators and boilers, industry, and agriculture.

We did not attempt to estimate the military fuel consumption. Military might be a large consumer of diesel in the country; however, none of the Russian experts was able to provide fuel estimates. We might assume that military consumption is spread throughout all sectors, but we cannot verify this assumption. We should note that the military

uses diesel with high sulfur content. Most of the military equipment is designed for high-sulfur fuel; Russia prohibits low-sulfur diesel for military goals. From 1 January 2015, the intergovernmental standard GOST 305-2013 requires the 2000 ppm sulfur content for the defense orders (Rosstandard, 2013).

## 4 On-road transportation

### 4.1 Activity data



**Registered fleet**


We use the complete vehicle registry containing information from about 49 million records to analyze on-road transport in Russia. The Russian analytical agency Avtostat provided the official registry with detailed vehicle information on fuel type and emissions standard (Avtostat, 2015b).

According to Avtostat data, there were 40.83 million passenger cars registered in Russia as of January 2015.

The share of diesel passenger cars was 4.2 % (Avtostat, 2015b). The popularity of diesel cars is growing in Russia; their fraction in new sales is 7-8 %. Russia does not have large-scale production of diesel passenger cars. Only 2 % out of 1.7 million diesel cars registered in Russia in 2014 were Russian models, while all the other diesel cars were imported or produced in Russia by foreign companies. Overall, foreign-make cars, both gasoline and diesel, constitute about 50 % of the passenger fleet.

Among the 3.96 million light commercial vehicles (LCVs) registered in 2014, 28 % used diesel. The share of diesel LCVs in new sales is also growing, and every other LCV sold in Russia is equipped with a diesel engine.

The heavy-duty truck fleet consists of 3.73 million vehicles. There were 2.32 million registered diesel trucks (62 % of the truck fleet). The fact that not all heavy-duty trucks use diesel plays an important role in emission calculations. Studies that assume that all heavy-duty trucks use diesel tend to overestimate their emissions.

In recent years, 98 % of new trucks run on diesel. Russian-make heavy-duty trucks constitute about two thirds of the diesel truck fleet. We grouped all diesel trucks into 4 groups depending on their weight: <7.5 t (35 % of the truck fleet), 7.5-12 t (19 %), 12-14 t (9 %) and above 14 t (37 %) (RAMR, 2012). This classification is consistent with the COPERT and NIIAT models.

There were 0.39 million buses registered in Russia in 2014. Forty-five percent of buses run on diesel. Russian

brands made up about two thirds of the diesel bus fleet. We group all diesel buses into 3 groups depending on their size: small buses (75 % of the bus fleet), medium (12 %), and large and extra-large (13 %).

**Distribution by emission standard**

Splitting the on-road fleet by fuel is important for emission calculations for two reasons. First, gasoline vehicles emit almost no BC. Hence, we do not analyze gasoline vehicles in this study. Second, the distribution of diesel vehicles

by emission standard is different from gasoline vehicles; overall, diesel vehicles are much newer than gasoline ones (Fig. 2).

For example, 91 % of gasoline trucks are Euro 0, while the share for diesel trucks without emission controls is 51 %. In addition, a significant number of diesel vehicles were imported in Russia, and as a result they meet higher emission standards.

Emission standards and fleet upgrades played an important role in emission reductions. For example, NIIAT estimated that from 2006, when Russia first had introduced emission standards, to 2011, PM emissions from on-road vehicles in the country dropped by 30 % (Donchenko, 2007, 2013). This happened even as the number of registered trucks and cars increased by 12 % and 36 %, respectively (GKS, 2014b).

**Active vehicles**





Russian experts point out that the official vehicle registry does not correctly reflect the number of vehicles on the roads (Donchenko, 2013, 2016; Avtostat, 2016, 2015c). A significant share of the fleet is very old: 28 % of cars and 49 % of LCVs are older than 10 years; 36 % of trucks and 23 % of buses are older than 20 years (Avtostat, 2015b). The fact that these vehicles are still registered does not mean that they are in working condition. For emission calculations we assess the "active fleet": the vehicles that are being used regularly.

To estimate the share of active vehicles, Avtostat used annual data from the Russian Union of Insurers about the number of insurance policies (stickers) issued. In other words, the total number of stickers issued is a good proxy for the active fleet because it is illegal to use vehicles without insurance stickers. According to Avtostat estimates, the share of active passenger cars is 76 % of the number of registered cars; for LCVs, buses, and trucks these shares are 80 %, 49 % and 64 %, respectively. Using the age distribution of diesel and gasoline vehicles, we calculated the

share of active diesel and gasoline fleet. Table 3 shows the summery of our calculations. Supplement Table S3 shows the number of active diesel vehicles by type and emission standard in Russia in 2014.

**Superemitters**

Superemitters should be represented in inventories because they are responsible for a large share of emissions. The concept of superemitters is not well-defined in the literature. The common approach is to define superemitters as

vehicles that have very high emissions compared to regular vehicles (sometimes they can be referred to as "high emitting vehicles"). In the vehicle testing studies, a cutoff level is used to determine the share of superemitters. For example, in Thailand, Subramanian et al. (2009) chose 4.7 g kg$^{-1}$ as the cutoff for all diesel superemitters in their Bangkok study. In Chile, Faiz et al. used the cutoff level of 7.5 g kg$^{-1}$ for buses in the Santiago study (Faiz et al., 1996).

For national emission inventories, the cutoff approach cannot be used for emission calculations. The commonly accepted approach is to define the share and emission factors of superemitters. As a result, this study uses assumptions about the share of superemitters in the diesel fleet to provide a more realistic emission inventory.

        The role of superemitters in emissions is very significant. For example, Ban-Weiss et al. (2009) measured emissions from 226 diesel trucks driving through a highway tunnel in California and found that 10 % of the highest-

emitting trucks were responsible for about 40 % of total BC from trucks. In Beijing, Wang et al. (2011) found that approximately 5 % of the trucks are responsible for 50 % of the BC emissions. In Europe, in a study of 139 individual vehicles of different types in Slovenia, Ježek et al. (2015) found that 25 % of the highest-emitting diesel vehicles produce 63 % of the BC emissions. Preble at al. found that 20 % of trucks emit 80 % of the BC emissions from the Port of Oakland truck fleet (Preble et al., 2015).

Superemitters emit a significant share of total emissions, but there is a limited number of studies on the share of diesel vehicles. There are various estimates of the share of superemitters in the fleet. For example, Subramanian et al (2009) estimated that the fraction of superemitters in the studied diesel fleet in Bangkok is 15 %. In their study of BC and PM emissions from 251 trucks in California, Ban-Weiss et al. (2009) found that about 13 % of the diesel fleet are superemitters. Bond et al. (2004) assumed with a high uncertainty that the share of superemitters for

countries "similar" to the United States is 5 %. A recent study by the California Air Resources Board shows that 8 %





of trucks were classified as high emitters (emitting over 5 % opacity) from a sample of over 1800 truck tests (CARB, 2015). We should note that US EPA no longer uses the concept of superemitters to estimate vehicle emissions (EPA, 2015).

There are no known studies on superemitters in Russia. Bond et al. (2004) assumed that the share of superemitters in Eastern Europe and the former Soviet Union is 10 %. This estimate also was used in other studies (Yan et al., 2011;Yan et al., 2014). This study uses the same assumption about the share of superemitters in Russia.

    We use a logistic function from (Yan et al., 2011) to represent the rate at which normal vehicles become superemitters (Eq. 2).

$$fr\,(s) = \frac{gain}{1+\exp[\alpha_{sup}(1-{}^{s}/_{L50\,sup})]} \qquad (2)$$

where $fr$ is the fractional rate at which normal vehicles become superemitters (fraction per year); $gain$ is the maximum rate of superemitter transition, $a_{sup}$ determines the slope of the transition curve with age, $s$ is vehicle age, and $L_{50,sup}$ is the vehicle life at which the rate becomes half the maximum.

    Since we already excluded retired (inactive) vehicles from the registry, we modified the $gain$ parameter in the formula to obtain the number of superemitters, which equal 10 % of the total active diesel fleet to be consistent with

previous studies (Yan et al., 2011;Yan et al., 2014;Bond et al., 2004). In this study, the parameters of the formula are as follows: $a_{sup} = 5.5$; $L_{50,sup} = 5.0$; $gain = 0.0162$.

    The share of superemitters in the fleet depends on the vehicle age. Using Formula 2 we calculated that this share is less than 1 % for vehicles less than 5 years old, close to 10 % among 10 year-old vehicles and 25 % for 20 year-old vehicles. Since the age distribution varies by vehicle type, using Formula 1 we calculated the fraction of

superemitters in the diesel fleet: 6.4 % for cars, 11.4 % for LCVs, 12.9 % for trucks and 10.2 % for buses. As mentioned above, the overall share of superemitters in the diesel fleet is 10 %.

    Using the information on diesel consumption by vehicle type and the percentage of superemitters in the fleet, we calculated that superemitters consumed 2100 t of diesel or 9.5 % of total diesel consumption by on-road vehicles. Based on Yan et al. (2011) we assume that PM emission factors for diesel on-road superemitters is 8.31 g kg$^{-1}$ for

older engine superemitters (Euro 0 and Euro 1) and 2.92 g kg$^{-1}$ for newer engine superemitters (Euro 2–Euro 5).

    Similarly, we assume that the share of superemitters in the off-road fleet is the same as in on-road one. Following Bond et al. we assume that the PM emission factor for off-road superemitters is 12 g kg$^{-1}$ and OC/BC share is 0.21 (Bond et al., 2004).

**Annual distance traveled**

The annual average distance traveled is one the most important parameters in the COPERT model. We use several sources to estimate the annual number of kilometers traveled by type of vehicles in Russia. NIIAT developed a methodology for assessing the residual value of vehicles based on their age and kilometers traveled (NIIAT, 1998). The methodology provides estimates of the annual average distance traveled by type of vehicles, country of production, and road type. In its emission calculation methodology (2008, 2012), NIIAT estimated the average

annual distance traveled for the total fleet. Avtostat conducted an extensive study of vehicle activity and estimated the average annual kilometers traveled by Russian and foreign-made cars. Avtostat also provided its estimates on



average kilometers traveled by LCVs, trucks, and buses. We use the Avtostat assumptions for emission calculations. Table S4 shows the assumptions on average of annual kilometers traveled in different models/methodologies. Supplement Table S5 provides details on our assumptions on annual kilometers traveled by vehicles by Euro class.

We assume that the average speed is 25 km h$^{-1}$ in cities, 40 km h$^{-1}$ on rural roads, and 90 km h$^{-1}$ on highways. We also assume that vehicles traveled 40 % of their annual distance on urban roads, 20 % on rural roads and 40 % on highways.

## 4.2    Emissions calculations

We calculate BC emissions from diesel on-road vehicles using the COPERT 4 model with NIIAT emission factors.

For our initial emission calculations with the COPERT model we exclude superemitters. We calculate emissions from superemitters using Formula 1 and added to COPERT results.

    Table 4 shows the results of emission calculations from active diesel vehicles. Heavy-duty trucks emitted 68 % all on-road diesel BC, while passenger cars emitted only 4 % of the emissions.

    The results show that superemitters emitted 8.28 Gg of BC or 40 % of all diesel on-road BC emissions. The role

of superemitters in emissions by type of vehicles varies: from 24 % for light-duty vehicles to 30 % for buses to 47 % heavy-duty trucks.

    The total BC emissions from on-road diesel vehicles are estimated at 20.51 Gg in 2014. Heavy-duty trucks emitted 70 % of all on-road diesel BC emissions. We also estimated that normal vehicles emitted 6.08 Gg of OC emissions, and superemitters produced additional 3.33 Gg of OC in 2014 (see Supplement Table S6 for details).

As we mentioned above, it is important to separate diesel vehicles from gasoline ones, exclude vehicles that are not in use, and factor in superemitters. If one assumed that all heavy-duty vehicles use diesel, BC emissions from trucks alone would be 36.79 Gg of BC, significantly overstating the total. Likewise, BC emissions from all registered diesel vehicles (as they appeared on the vehicle registry) are 27.31 Gg. Emissions from the adjusted fleet without accounting for superemitters would be 14.85 Gg (See Supplement Tables S7-S9 for details).

# 5    Off-road diesel sources

## 5.1    Rail

The total length of railroads is 86 000 km and about 60 % of them are electrified (GKS, 2014a). Given the size of the country, the density of railroads is low compared to other European countries. Rail cargo turnover was 2301 billion tkm in 2014 which is almost ten times larger than that of road transport (247 billion tkm). In 2013, diesel

locomotives carried almost 15 % of all rail cargo (GKS, 2014a).

    The Russian Railway Company (RZhD, based on the Russian acronym) is the largest owner of diesel locomotives in the country. RZhD owned 10 400 electric locomotives and 10 200 diesel locomotives in 2013, including 3500 line haul and 6100 shunting locomotives (Balabin and Evpakov, 2013). In addition to RZhD's stock, large industrial companies also own about 12 000 locomotives to form trains. In 2012, RZhD started using Euro 3

diesel (350 ppm) for its diesel locomotives (RZhD, 2013).



The locomotive fleet is old: about 50 % of long-line haul locomotives are more than 15 years old. Diesel locomotives in Russia have no emission controls. The EEA guidebook presents the emission factors for diesel locomotives based on average European fleet (1.37 g kg$^{-1}$). Given that the Russian locomotives are older than those in Europe, we use the emission factor from Yan (2014) for locomotives without emission controls. Thus, we assume that the PM emission factor for diesel locomotives in Russia is 4.62 g kg$^{-1}$.

### 5.2 Domestic navigation and fishing

Domestic navigation and fishing represent different economic sectors but use similar combustion technologies. Liquid bulk ships, dry cargo carriers, and container ships mainly use heavy bunker fuel oil, while passenger ships, fishing boats, and tugs use diesel. Diesel ships tend to be smaller than those using bunker fuel oil. Almost all ships use diesel during maneuvering and while docked at shore. As a result, emissions from domestic navigation and fishing are presented in the same category.

Russia is a large marine state with the third longest coastline in the world. There are 67 sea ports in Russia although only a few are ice-free in winter. The largest areas of maritime activity are the Baltic Sea, Black Sea region, and the Far East. The Arctic region accounts for 5.6 % of cargo turnover, but maritime activity there is rapidly expending, given the increasingly ice-free Northern Sea Route.

Most marine and fishing vessels in Russia are old. For example, over 70 % of river and lake vessels are older than 25 years (Mintrans, 2015). Similarly, over 80 % of fishing ships are over 20 years old (WCIOM, 2015).

The cargo fleet has been shrinking: there were 3830 sea-going vessels in 2000, 3514 in 2005, and 2712 in 2014. The number of river vessels decreased from 31 800 in 2000 to 21 800 in 2014 (GKS, 2015b).

We estimate that ships use 526 kt of diesel in 2014. This does not include military consumption, which could be very significant. Since Russia does not have any emission standards for ships, we assume that ships have no emission controls.

### 5.3 Agriculture

According to estimates from the Ministry of Agriculture, in 2014 agricultural companies owned 420 000 agricultural tractors, 153 000 harvesters and 22 000 other motor vehicles (Ministry of Agriculture, 2015). The agricultural fleet has been shrinking in Russia (Ministry of Agriculture, 2015). For example, in 2014 there were 15 000 fewer tractors than in 2013. In 2014 the retirement rate for tractors was 5.1 % while the replacement rate was 3.2 % (GKS, 2015a).

Tractors produced in Russia, Belarus, and Ukraine constitute over 90 % of the tractor fleet, and the majority of tractors are over 10 years old (Ministry of Agriculture, 2015). As a result, the availability of emission controls is very limited; moreover, until recently, there were no emission standards for agricultural vehicles in Russia. It is possible that emissions will drop in the future because of new emission standards for agricultural vehicles in Russia.

Since Russia has limited production of agricultural tractors (about 3 % of new sales), foreign-made agricultural machinery dominates in new sales (Agroinfo, 2015). The share of used tractors in the total imports was 20 % in 2014. We assume that used tractors imported from Western countries have emission controls; however, their share in the total agricultural fleet is very small.



We assume that the distribution by emission standard is as follows: 95 % is Stage 0 (without emission controls) and 5 % meets Stage 2 standards.

### 5.4 Industry

**Mining**

The mining sector consumes about half of the industrial diesel. Russia is a major mineral and coal producer. Russia produced 357 Mt of coal in 2014; 65 % of coal was produced in open-pit mines (EMIS, 2014;GKS, 2015b). Open-pit mining is widespread in Russia due to its relatively lower production costs. Russia is also a leading global producer of many other mined commodities, including aluminum, copper, iron ore, lead, and nickel, among others (USGS, 2016).

Mining trucks consume 70–80 % of the diesel at open pit mines both due to their large engines and the fact that mining operations continue nonstop. On average each truck operates well over 6300 h yr$^{-1}$ (Mining Magazine, 2007). The Belarusian company BELAZ supplies the majority of the largest mining trucks (Petrovich et al., 2013) and most BELAZ trucks are equipped with Cummins and MTU engines. The average life of mining trucks is short: BELAZ trucks operate 5–7 years (Zvonar, 2010) while Caterpillar mining trucks can operate for 9-12 years 365 (Anistratov, 2013).

Russia has no emission standards for off-road mining vehicles, and as a result Western companies can supply engines without emission controls. For example, about 88 % of Cummins engines in Russia have no controls, and the remaining 12 % meet US EPA Tier 1 requirements (Mueller, 2014). A small population of Caterpillar and Komatsu trucks also meets Tier 1 or Tier 2 requirements.

The role of the mining industry in diesel consumption and BC emissions is especially important in the Russian Arctic. For example, in their study of BC emissions in the Arctic, Evans et al. (2015) found that the mining industry emits about 70 % of all diesel BC emission in the Murmansk Region. A second study on BC emissions in the Russian Arctic found that the mining industry emits 80 % of BC emissions from combustion in Russia's Arctic zone (Morozova, 2015).

We assume that 88 % of engines in the mining industry are Tier 0 (1991–Stage 1) and 12 % are Tier 1 (Stage 1).

**Construction**

The construction industry is an important sector of the Russian economy. In 2013, 5.7 million people were employed in construction (8 % of the labor force). Over 226 000 construction companies worked in Russia in 2014 (GKS, 2015b), meaning that most of them are small businesses.

The construction industry uses more varieties of diesel engines than any other sector of the economy. Most of construction equipment is old and lacks emission controls. About 30-50 % of these excavators, loaders, bulldozers, and graders have reached their end of useful lifetime (Rosstat, 2014). Though up to 60 % of construction machinery is imported depending on type of vehicles, they do not necessarily have emission controls. We assume that the distribution by emission standard is as follows: 90 % of construction machinery is Stage 0 (without emission 385 controls) and 10 % have some emission controls.



**Other industry**

Other types of industries that use diesel include production of iron, steel, non-ferrous metals, chemicals, machinery, food, paper, wood products, textiles, and other types of goods. There is a huge variety of diesel machinery and equipment in these industries. We assume that this is primarily heavy industrial equipment with no emission
controls.

### 5.5     Diesel generators

About 60 % of Russia's territory is not connected to the centralized electricity grid (Suslov, 2012). Twenty million people live in these off-grid areas, which include cities, towns, and villages (Zatopliaev and Redko, 2004). Stationary diesel generators produce electricity in small isolated grids in remote locations.
About 47 000 diesel generators provide electricity; 12 000 of these are in the northern part of the country. The typical generator power capacity varies from 100 kW to 3.5 MW. At the beginning of the 2000s, the installed capacity of diesel generators was about 17 million kW or 8 % of the total installed capacity in Russia (Minenergo, 2012). According to the Russian Statistical Service, large diesel power stations generated 4500 GWh electricity in 2014 (Fedstat, 2015h).
In addition to electricity generation, diesel can be used to produce heat. Diesel boilers and heat pumps are used in areas without centralized district heating. The process of external combustion in boilers is quite different from that in diesel engines and as a result PM emission factors and BC/PM ratios for external combustion are lower than those from internal combustion engines (Bond et al., 2004).
In 2014 diesel generators used 1.034 Mt of diesel, and heat plants used an additional 315 kt.

### 6     Results of BC emission calculations
Table 5 presents the results of emission calculations from off-road diesel sources in Russia as well as the total for on-road transportation in 2014. We estimate that all off-road sources emitted 22.9 Gg of BC and 5.3 Gg of OC. The largest emission contributors in the off-road sector are industry, locomotives, and diesel generators. This study also includes superemitters in the off-road diesel fleet. The role of superemitters in the off-road fleet is less important
than for the on-road fleet due to differences in emission factors between normal engines and superemitters. We estimate that off-road superemitters are responsible for 21% of BC emissions from off-road sources.
Rail is the largest source of off-road BC emissions because of outdated equipment and high emission factors, as well as the extensive use of diesel locomotives in off-grid parts of Russia. Industry is a large source because of the diversity of small uses without emission controls. Diesel generators without emissions controls produced more BC
emissions than the mining industry because of lack of emission controls and larger emission factors.
These results show that off-road diesel sources emit 57 % of total diesel BC in Russia. These high levels of emissions from off-road sources are a result of limited use of emission control technologies, a function both of the equipment age and the lack of regulations for new equipment. This contrasts with emissions from on-road vehicles,



where standards were introduced a decade ago, and emissions have subsequently dropped. While consuming 70 % of the diesel fuel in the country, on-road vehicles produced 43 % of BC emissions in 2014.

**7        Comparison with other studies**

There have been several studies looking at BC emissions in Russia across a range of sectors, but to date, the majority of these studies use fuel-based, mass balance approaches to calculating emissions. In previous studies, emissions from on-road transport were estimated based on the number of registered vehicles or in the best case on

vehicles separated into a few emission standards. A limited number of studies have assessed the existence of control technologies and other detailed, real-world activity data. None of the studies accounted for superemitters in the fleet.

Table 6 below shows the result of several previous studies covering total anthropogenic BC emissions, emissions from transport or all diesel sources.

There are two wide categories of studies on Russian BC emissions. Emission estimates in the first category are

based on fuel consumption, use global or regional emission factors, and mostly do not use Russian activity data. For example Bond et al. (2004) combined fuel consumption data and application of combustion technologies and emission controls (Bond et al., 2004;Sarofim et al., 2009). Lamarque et al. (2010), updating the Bond data (2004), estimated that diesel engines are likely to be the fourth largest source of BC emissions in Russia after residential/domestic sources, forest fires, and industry (EPA, 2012b; Lamarque et al., 2010). The International

Institute for Applied Systems Analysis (IIASA) uses the Regional Air Pollution INformation and Simulation (RAINS) model and ECLIPSE (Evaluating the Climate and Air Quality Impacts of Short-Lived Pollutants) model to estimate PM and BC emissions. BC emissions from transport were estimated at 52 Gg in 2010 (Sand et al., 2016).

The second category of Russian BC studies is based on Russian activity data. They use Russia-specific emission factors for on-road transport and/or bottom-up fuel consumption data. Some of these studies were completed in the

framework of BC mitigation efforts in the Arctic. Russia submitted its first report to the Arctic Council on BC and methane emissions reductions in 2015. The total Russia-wide BC emissions were estimated at 358.5 Gg in 2013 (MNRE, 2015). Transport was not estimated as a significant source of BC emissions (7.7 Gg or just 2 % of the emissions). An international group of scientists led by the US Department of Energy estimated BC emissions in the Russian Arctic and in Russia from anthropogenic sources (Huang et al., 2015). Drawing on local Russian

information, Russian BC emissions were estimated at 224 Gg in 2010. Using data from vehicle registry, BC emissions from transport were estimated at 45.3 Gg (Huang et al., 2015). The authors assumed that all registered vehicles are being actively used and all vehicles use diesel fuel. The paper does not present the vehicular emissions by vehicle type or emission standard.

NIIAT estimated that Russian on-road vehicles emitted 53.9 Gg PM in 2006 (Donchenko, 2007) and 38.5 Gg

PM in 2011 (Donchenko, 2013). NIIAT used the Russian PM emission factors. Until recently, it had not calculated BC emissions. NIIAT used detailed information about the number of diesel vehicles and adjusted the registry to reflect the share of the active fleet. NIIAT does not account for superemitters in emission calculations.





Evans et al. (2015) used the IEA diesel data and estimated BC emissions from diesel transport and all diesel sources in Russia. Using fuel-based emission factors from the EEA emission guidebook and NIIAT emission
factors, they estimated that BC emissions from on-road transport were about 20.0 Gg (Evans et al., 2015).

We can conclude that the results of the emission calculations presented in the current study are close to those studies which used detailed Russian activity data (number of active diesel vehicles, annual average distance traveled, Russian emission factors). The advantages of the current study are that we present BC emissions from on-road transport by vehicle types and emission standards, factor in superemitters, and also present OC emissions from on-
road vehicles and off-road diesel sources.

**8        Uncertainty**

There are two major sources of uncertainty in BC emission inventories: 1) emission factors and 2) activity data. Emission factor uncertainty includes uncertainties in PM emission factors for normal vehicles and superemitters, and BC/PM speciation ratios. NIIAT does not report the uncertainty in PM emission factors for on-road vehicles. In
COPERT, the uncertainty for PM emission factors is estimated to be 20–30 % (Kouridis et al., 2010). Uncertainty in PM emission factors for off-road sources is 30-60 % for agricultural vehicles, 25-50 % for ships, and an order of magnitude for industry (EEA, 2013).

Uncertainty in $BC/PM_{2.5}$ speciation ratios for on-road vehicles is 5–10 % for light-duty vehicles and 20 % for heavy-duty engines. The speciation ratio uncertainty for off-road diesel sources is 20 % (EEA, 2013). The speciation
ratios are not a major source of uncertainty in emission inventories for diesel BC sources.

Activity data also present uncertainties because of uncertainties in underlying surveys or estimation methodologies. This includes data on fuel consumption by sector, distribution by vehicle type, annual number of kilometers traveled, and assumptions about emission controls. Fuel data differ by a lot, as the average annual distance traveled. The distribution by vehicle type and controls for on-road transport is less uncertain. Off-road
uncertainty on emission controls is larger because it is possible that more emission controls exist than we assume. We use several approaches to minimize uncertainties in the activity data. These include multiple approaches to data collection, cross-checks with the literature, and expert judgments.

Our data on fuel consumption are based on bottom-up calculations that are close to the Russian official statistics. While there is uncertainty in the distribution between economic sectors, the total domestic diesel supply is
well-determined.

Assumptions on emission controls do not significantly contribute to uncertainties in emissions because we assume that about 90–95 % of off-road diesel engines have no emission controls. However, real emission factors for Russian diesel sources are not well understood. For the on-road vehicles, the distribution by emission standards is well determined based on the registry. The key uncertainty here is the assumption of the share of active vehicles.
Another source of uncertainty is the share of superemitters in the on-road and off-road fleet. For emission calculations, we assume that the share of superemitters is 10 %. This number is to some extent arbitrary because it was determined based on a small number of studies. In addition, heavy-duty trucks are designed to meet emissions limits up to a specified maximum loading; overloading can significantly increase the share of high-emitting vehicles



(World Bank, 2014). There is evidence that Russian drivers tend to overload their trucks, especially on the long-haul
routes to save time and increase their short-term profit. According to the Russian Federal Road Agency, 30–40 % of
heavy-duty trucks are overloaded on average by 45 % (Avtodor, 2015). As a result the share of superemitters in the
truck fleet might be much higher.

## 9 Conclusions

In this paper, we estimate BC and OC emissions from diesel sources in Russia. We use detailed vehicle registry
containing information from about 49 million records to analyze on-road transport in Russia. We separate diesel
vehicles from gasoline ones, estimate the share of active vehicles in the fleet, use detailed information on
distribution by vehicles types and emission standards, and use Russia-specific emission factors for emission
calculations. This study also factors in the role of superemitters in BC emissions from on-road diesel vehicles.
Emissions from on-road diesel vehicles are estimated at 20.5 Gg of BC and 9.4 Gg of OC in 2014. Heavy-duty
trucks emitted 70 % of BC, while diesel passenger cars emitted only 4 % due to their small share in the total
passenger fleet and availability of emission controls. Assuming that the share of superemitters is 10 % in the on-road
diesel fleet, we estimate that these high-emitting vehicles are responsible for 40 % of all BC emissions from on-road
vehicles. Under this assumption, the role of superemitters in emissions by vehicle type varies from 24 % for light-
duty vehicles to 30 % for buses to 47 % for heavy-duty trucks. This reflects the fact that the truck fleet is much older
than the fleet of diesel passenger cars.

We also estimate BC emissions from off-road diesel sources including diesel locomotives, ships, off-road
engines in industry, construction and agriculture, and from diesel generators. We estimate that off-road diesel
sources emitted 27.7 Gg of BC in Russia in 2014. Off-road diesel sources also emitted 6.5 Gg of OC. Stationary
engines in industry are the largest source of off-road BC emissions followed by locomotives and diesel generators.
Off-road diesel sources emitted 57 % of all diesel BC emissions. Off-road superemitters emitted 21% of emissions
from off-road diesel sources.

The total emissions from diesel sources in Russia are estimated to be 48.2 Gg of BC and 15.9 Gg of OC in
2014.

### Acknowledgments

The authors are grateful for research support provided by the US Environmental Protection Agency, Office of
International and Tribal Affairs (under the inter-agency agreement DW-089924383) and the US Department of
State. Battelle Memorial Institute operates the Pacific Northwest National Laboratory for the US Department of
Energy under contract DE-AC05-76RL01831. We thank the members of the Technical Steering Group for their
helpful comments and suggestions. The views and opinions expressed in this paper are those of the authors alone.






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



**Table 1.** PM$_{2.5}$ emission factors and BC/PM ratios for off-road diesel sources.

| Sector | Emission controls | PM$_{2.5}$, g kg$^{-1}$ | Source | BC/PM$_{2.5}$ ratio | Source |
|---|---|---|---|---|---|
| Diesel generators | No control | 6.0 | Bond et al. (2004) | 0.66 | Bond et al. (2004) |
| Heat plants | No control | 0.25 | Bond et al. (2004) | 0.29 | Bond et al. (2004) |
| Industry (stationary engines) | No control | 4.308 | EEA (2013), 1.A.4.,Table 3-2 | 0.55 | EEA (2013), 1.A.4.,Table D3 |
| | 1991-Stage I | 3.551 | EEA (2013), 1.A.4.,Table 3-2 | 0.55 | EEA (2013), 1.A.4.,Table D3 |
| | Stage II | 1.031 | EEA (2013), 1.A.4.,Table 3-2 | 0.80 | EEA (2013), 1.A.4.,Table D3 |
| Construction | No control | 4.308 | EEA (2013), 1.A.4.,Table 3-2 | 0.55 | EEA (2013), 1.A.4.,Table D3 |
| | 1991-Stage I | 3.551 | EEA (2013), 1.A.4.,Table 3-2 | 0.55 | EEA (2013), 1.A.4.,Table D3 |
| Rail | No control | 4.62 | Yan et al. (2014) | 0.65 | EEA (2013), 1.A.3.c.,Table A.1 |
| Ships | No control | 1.4 | EEA (2013), 1.A.3.d.,Table 3-2 | 0.31 | EEA (2013), 1.A.3.d., Table 3-2 |
| Agriculture | No control | 3.755 | EEA (2013), 1.A.4.c.ii, Table 3-2 | 0.55 | EEA (2013), 1.A.4.c ii, Table D3 |
| | 1991-Stage I | 1.644 | EEA (2013), 1.A.4.c ii, Table 3-2 | 0.54 | EEA (2013), 1.A.4.c ii, Table D3 |
| | Stage I | 0.832 | EEA (2013), 1.A.4.c ii, Table 3-2 | 0.79 | EEA (2013), 1.A.4.c ii, Table D3 |
| | Stage II | 0.627 | EEA (2013), 1.A.4.c ii, Table 3-2 | 0.77 | EEA (2013), 1.A.4.c ii, Table D3 |



**Table 2.** Diesel consumption by sector in 2014 (kt).

| Diesel consumption | 2013 | 2014 |
|---|---|---|
| **Domestic supply** | **30 350** | **31 991** |
| Transformation processes | 1211 | 1349 |
|   Electricity plants | 945 | 1034 |
|   Heat plants | 266 | 315 |
| Energy industry own use (coal, oil, gas) | 173 | 236 |
| **Final consumption** | **28 966** | **30 406** |
| Industry | 2785 | 3279 |
|   Mining | 1281 | 1509 |
|   Other industry | 1504 | 1771 |
| Construction | 438 | 414 |
| Transport | 23 993 | 24 970 |
|   Rail | 2337 | 2261 |
|   Road | 21 066 | 22 189 |
|   Domestic navigation | 423 | 372 |
|   Other transport | 167 | 148 |
| Agriculture and fishing | 1749 | 1711 |
|   Agriculture | 1592 | 1557 |
|   Fishing | 157 | 154 |
| Other | 31 | 32 |

Source: (CENEf, 2016)





**Table 3.** Percentage of active diesel and gasoline vehicles.

| | Share of all active vehicles (Avtostat) | Share of active diesel vehicles (calculated) | Share of active gasoline vehicles (calculated) |
|---|---|---|---|
| Cars | 76 % | 86 % | 75 % |
| LCVs | 80 % | 84 % | 67 % |
| Trucks | 49 % | 57 % | 41 % |
| Buses | 64 % | 70 % | 63 % |






**Table 4.** BC emissions from active on-road diesel vehicles in 2014, (Gg).

|  | Cars | LCVs | Trucks | Buses | Total |
|---|---|---|---|---|---|
| Euro 0 | 0.12 | 0.82 | 4.07 | 0.23 | 5.24 |
| Euro 1 | 0.05 | 0.31 | 0.38 | 0.10 | 0.85 |
| Euro 2 | 0.11 | 0.46 | 1.19 | 0.21 | 1.98 |
| Euro 3 | 0.11 | 0.52 | 1.55 | 0.33 | 2.51 |
| Euro 4 | 0.21 | 0.93 | 0.34 | 0.04 | 1.52 |
| Euro 5 | 0 | 0 | 0.11 | 0 | 0.12 |
| Total Euro 0-5 | 0.62 | 3.05 | 7.65 | 0.91 | 12.23 |
| Superemitters | 0.20 | 0.94 | 6.70 | 0.44 | 8.28 |
| TOTAL | 0.82 | 3.99 | 14.35 | 1.35 | 20.51 |





**Table 5.** BC and OC emissions in Russia in 2014 (Gg).

| Sector | BC | OC |
|---|---|---|
| On-road vehicles | 20.5 | 9.4 |
| Rail | 7.6 | 1.5 |
| Ships | 0.6 | 0.1 |
| Mining | 4.2 | 1.3 |
| Other industry | 5.2 | 1.0 |
| Diesel generators | 4.5 | 0.9 |
| Agriculture | 4.0 | 1.2 |
| Construction | 1.1 | 0.3 |
| Other sectors | 0.4 | 0.1 |
| Total | 48.2 | 15.9 |





**Table 6.** The results of BC emission studies for Russia.

| Study | Base year | Emission source categories | BC emissions (Gg) |
|---|---|---|---|
| Bond et al. (2004) | 1996 | Agricultural burning, industry, open fire, power generation, residential biofuels, road-transport, off-road transport | 200 |
| Lamarque et al. (2010) | 2000 | Residential/domestic sources, forest fires, industry, diesel engines | 360 |
| | | Diesel transport (including aircrafts and marine shipping) | 32 |
| IIASA ECLIPSE dataset (Sand et al., 2016) | 2010 | All anthropogenic emissions (domestic, energy/industrial/waste, transport , agricultural fires, gas flaring) | 182 |
| | | Transport | 52 |
| Huang et al. (2015) | 2010 | Flaring, residential, transport, industry, and power plants | 224 |
| | | On-road transport | 45 |
| Russia's National Report to the Arctic Council (MNRE,2015) | 2013 | All anthropogenic emissions (agriculture, industry, transport, services) | 359 |
| | | Transport | 8 Gg |
| Donchenko (2006) | 2006 | On-road transport | 54 Gg PM (29 Gg BC)* |
| Donchenko (2013) | 2011 | On-road transport | 39 Gg PM (20 Gg BC)* |
| Evans et al. (2015) | 2010 | All diesel sources | 46 |
| | | On-road diesel transport | 20 |
| This study | 2014 | All diesel sources | 43 |
| | | On-road diesel transport | 21 |

\* - Assuming that BC/PM speciation ratio for on-road transport is 0.53 (EEA, 2013)





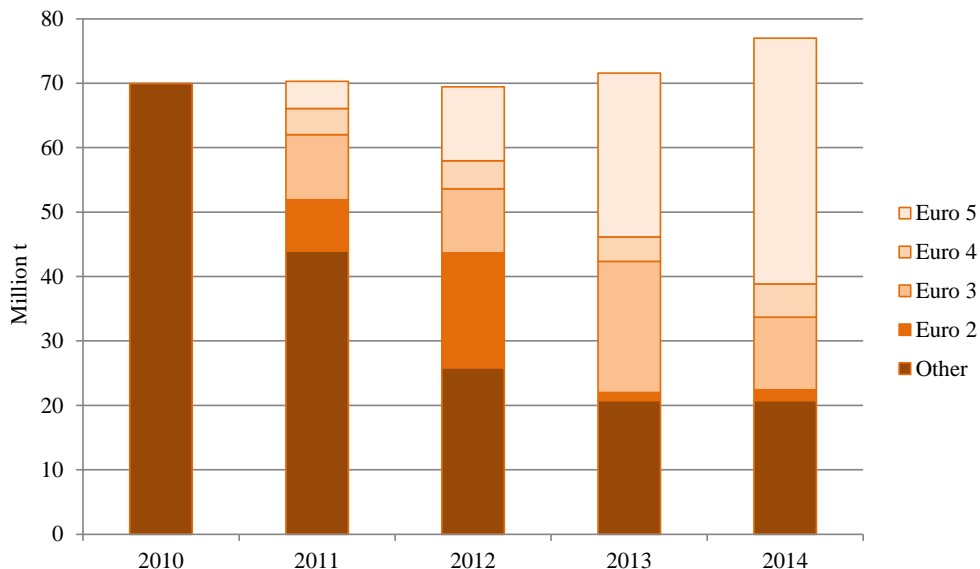

**Figure 1.** Production of diesel fuel by Euro class in Russia, 2010-2014, Mt.

Source: (Fedstat, 2015g).






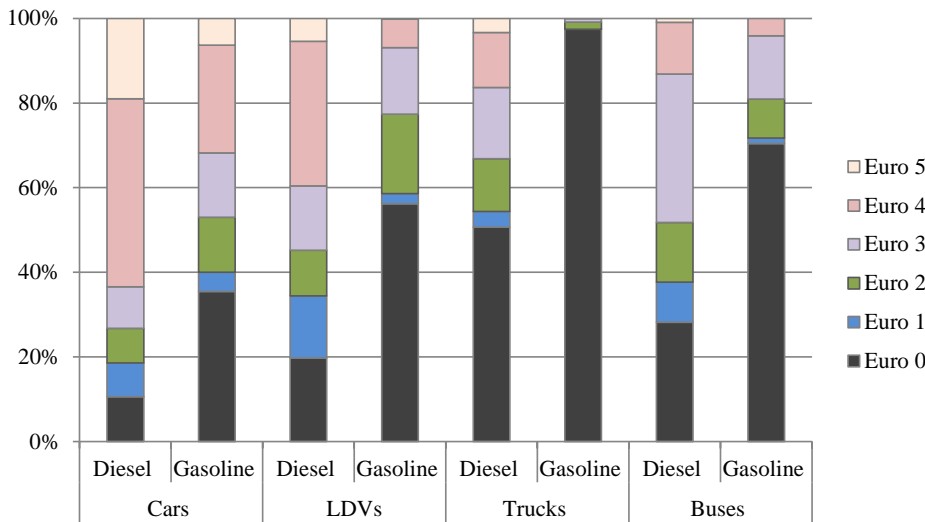

**Figure 2.** The distribution of diesel and gasoline vehicles by emission standard.

Source: (Avtostat, 2015b)
