# Peer review of "Russia's black carbon emissions: focus on diesel sources"

_Atmospheric Chemistry and Physics, 2016_

## Referee Comment (RC1) · Anonymous Referee #1 · 12 Jul 2016

This paper adds important information to the effort to understand emissions of key pollutants in Russia and the Arctic. The paper is well-organized and lays out step-by-step the process the authors used to calculate the emission changes. The methodologies appear sound and assumptions are largely explained in a satisfactory way. The references are extensive and robust and the supplemental tables and charts are helpful to guiding the reader's understanding of the material.

I would recommend the authors consider the following minor revisions before the article is published (in order of appearance):

1. The health information on p. 1 should be updated to reflect the most recently available health impact information (GBD 2013 was published in early 2016). 2. On p. 4, the authors discuss the movement to low sulfur fuel, but omit an important piece of

information – low sulfur fuel is necessary for operation of the most effective emission control devices on diesel vehicles. This would be additional context that would be helpful to add. 3. If the information is available, it would be informative to include the share of industry that is made up of small businesses rather than the number of people employed to give a better indication of the impact of the lack of reporting. (p. 5, lines 140-150) 4. Authors should consider defining "fuel balance approach" at first mention rather than later in discussion in case readers are not familiar with this methodology. (p. 5, line 155) 5. It is not clear why there is discussion of gasoline vehicles in the section of distribution by emission standards. The authors note that they produce almost no BC, so could easily be left out or addressed with a sentence explaining that they produce almost no BC. If there is a reason the comparison of emission standards is important to the discussion, this should be clarified. (p. 6, lines 198-208) 6. For logical flow, authors should consider moving the active vehicles section (p. 7, lines 210-221) to immediately following the registered fleet section. 7. The assumptions for speed and type of road traveled would benefit from further explanation (i.e., are these based on standard speeds/distribution of roads in Russia?) (p. 9, line 285) 8. The assumptions for controls on agricultural vehicles would benefit from further explanation (i.e., who no Stage 1?) (p. 11, line 351) 9. For logical flow, recommend moving the paragraph on uncertainty regarding BC/PM ratios after the paragraph on activity data (p. 14). It appears that the BC/PM ratios are NOT a major source of uncertainty, so it would make more sense for the reader if this is discussed after the two major sources of uncertainty that are identified.

---

## Author Comment (AC1) · 27 Jul 2016

Authors' comments on Review #1 Atmos. Chem. Phys. Discuss., doi:10.5194/acp-2016-475-RC1, 2016

Russia's black carbon emissions: focus on diesel sources" by N. Kholod et al. Minor revisions

1. The health information on p. 1 should be updated to reflect the most recently available health impact information (GBD 2013 was published in early 2016). Thank you for the suggestion. We updated the text as follows (Lines 36-37 of the updated text): "According to the Global Burden of Disease study, ambient PM pollution caused over 100,000 premature deaths in Russia in 2013 (GBD, 2016)." GBD: The global burden of disease study 2013, Institute for Health Metrics and Evaluation, Seattle, WA.

[Figure]

Available at http://ghdx.healthdata.org/gbd-data-tool (Last access: 12 July), 2016.

2. On p. 4, the authors discuss the movement to low sulfur fuel, but omit an important piece of information – low sulfur fuel is necessary for operation of the most effective emission control devices on diesel vehicles. This would be additional context that would be helpful to add. Thank you. We updated the text as follows (L 118-122): "Sulfur content of diesel fuel is an important factor in emission reductions. Diesel with high sulfur content (measured in parts per million or ppm) can destroy emission control devices, such as particulate filters. Availability of low sulfur diesel is an important prerequisite for the introduction of more stringent vehicle emission standards."

3. If the information is available, it would be informative to include the share of industry that is made up of small businesses rather than the number of people employed to give a better indication of the impact of the lack of reporting. (p. 5, lines 140-150). Thank you for pointing this out. Small businesses are not required to submit this information, yet they employ 11 million people (Fedstat, 2015f, a, b, e) and produced more than 20% of goods and services (GKS, 2015a). GKS: Russian Statistical Yearbook 2015. Federal Statistics Service of the Russian Federation. Available at http://www.gks.ru/free_doc/doc_2015/year/year15.rar (last access: 26 February 2016), 2015a.

4. Authors should consider defining "fuel balance approach" at first mention rather than later in discussion in case readers are not familiar with this methodology. (p. 5, line 155) We added to the text (Lines: 160-161): However, their assessments both have their limitations because they do not use a fuel balance approach; in other words they do not match diesel consumption by on-road vehicles and off-road engines with the production of diesel fuel in the country.

5. It is not clear why there is discussion of gasoline vehicles in the section of distribution by emission standards. The authors note that they produce almost no BC, so could easily be left out or addressed with a sentence explaining that they produce almost

no BC. If there is a reason the comparison of emission standards is important to the discussion, this should be clarified. (p. 6, lines 198-208) Thank you for this suggestion. We deleted the discussion about gasoline vehicles. The updated text (L 216-219): "Splitting the on-road fleet by fuel is important because gasoline vehicles emit almost no BC. Hence, we do not analyze gasoline vehicles in this study. Figure 2 shows the distribution of diesel vehicles by emission standard."

6. For logical flow, authors should consider moving the active vehicles section (p. 7, lines 210-221) to immediately following the registered fleet section. Thank you, we moved the Active Vehicle section after the Registered Fleet section. (For easy reading, the changes were not tracked).

7. The assumptions for speed and type of road traveled would benefit from further explanation (i.e., are these based on standard speeds/distribution of roads in Russia?) (p. 9, line 285) The assumption on the speed in cities is based on actual speed in Moscow and other large cities. The assumptions on the average speed on rural roads and highways are based on maximum allowable (standard) speed on these types of roads in Russia. The updated text (L 291-294): "The assumption on the speed in cities is based on actual speed in Moscow and other large cities. The assumptions on the average speed on rural roads and highways are based on maximum allowable (standard) speed on these types of roads in Russia. The share of vehicle-kilometers traveled on urban roads is taken from ICCT Roadmap model."

8. The assumptions for controls on agricultural vehicles would benefit from further explanation (i.e., who no Stage 1?) (p. 11, line 351) We assume that 95% of agricultural fleet has no emission controls and 5% meets Stage 2 standards. Only imported used tractors may have emission controls. In the European Union, Stage 1 emission standard was implemented in 1999 and Stage 2 implemented from 2001 to 2004. Stage 1 tractors might be too old for importing them to Russia so we assume that all imported tractors with emission controls meet Stage 2 standard. Stage 3 standards were phased in from 2006 to 2013, and these tractors are too new to be sold as used in 2014. We

updated the text as follow (L 356-357): "We assume that used tractors imported from Western countries have emission controls; however, their share in the total agricultural fleet is very small (no more than 5%).

9. For logical flow, recommend moving the paragraph on uncertainty regarding BC/PM ratios after the paragraph on activity data (p. 14). It appears that the BC/PM ratios are NOT a major source of uncertainty, so it would make more sense for the reader if this is discussed after the two major sources of uncertainty that are identified. Thank you for the suggestion. We moved the paragraph (L 485-487).

Thank you for the useful comments and suggestions!

---

## Referee Comment (RC2) · Anonymous Referee #2 · 13 Aug 2016

Overall, this is a good effort at building a better emissions inventory for Russian on-road and off-road diesel use (though excluding military diesel usage.) I am glad the authors account for super-emitters.

However, I have some concerns regarding the uncertainty propagation, the super-emitter fraction, and relatively unexplained emission factor data. Once these are clarified, this manuscript should be acceptable for publication.

One final concern is that the current submission has no explanation of differences between this paper, and the on-road BC emissions estimate published earlier by the first author (Kholod and Evans, http://dx.doi.org/10.1016/j.envsci.2015.10.017 ) While the current paper is more detailed, the bottom line figure appears the same - in 2015, Figure 1 of Kholod and Evans shows 20,000 tons of BC from on-road Russian sources,

similar to the current paper. Maybe the complicated model of the current submission is not needed?!

Detailed comments: Lines 181-184: authors claim "Russia does not have large-scale production of diesel passenger cars", but also say 98% of diesel cars were either imported or produced in Russia by foreign companies. Not sure how this affects the overall emissions, but what fraction of on-road diesel cars are made in Russia by Russian or non-Russian companies? (Also, since this paper focuses on diesel cars, the line about "foreign-make cars, both gasoline and diesel" is superfluous.)

Lines 203-204: what higher emissions standards do imported diesel vehicles meet? Euro 6? Or were imported vehicles always produced to meet a higher standard than necessary for Russia?

Lines 240-251: The authors rely on the Bond et al. (2004) assumption of super-emitter fraction as 10%, even though they cite several more recent studies that show super-emitters can be as high as 13-15% of the fleet, even in California. Given the lack of studies in Russia, and the authors' literature survey of the Russian fleet (36% of trucks and 23% of buses older than 20 years), using the old Bond et al. (2004) assumption will likely bias their emissions inventory low as the authors acknowledge at the end. The authors should investigate the sensitivity of their results to this fraction, and perhaps try higher values (15-30%) for the super-emitter fraction?

Lines 286-287: What is the basis for their assumption of 40-20-40 on urban roads, rural roads, and highways?

The authors use NIIAT data for on-road emission factors, but the actual source of that data is not clear – are these based on measurements or on estimates based on emissions standards? The authors present the data used; a brief explanation of the source methodology will be helpful, since these NIIAT publications do not appear to be easily accessible online. (I checked their website. I don't know Russian.)

The conclusions should note that the results exclude military diesel usage emissions; in particular, these could be large sources of sulfate PM, and possibly also BC.

While the authors present a comprehensive list of potential uncertainties with their emissions inventory estimate, they don't propagate the uncertainties through, which would be helpful. From their list, it appears the emission factors could produce uncertainties of +/-30% or so, while the bias due to low super-emitter fraction (10% when 15-20% might be more appropriate) could increase the overall BC estimate by as much as 40%!

---

## Author Response (AR1)

Atmos. Chem. Phys. Discuss.,
doi:10.5194/acp-2016-475
**Russia's black carbon emissions: focus on diesel sources" by N. Kholod et al.**

**Authors' response**

**Authors' comments on Review #1**

Minor revisions

**Comment 1.** The health information on p. 1 should be updated to reflect the most recently available health impact information (GBD 2013 was published in early 2016).

Thank you for the suggestion. We updated the text as follows (Lines 36-37 of the updated text):

"According to the Global Burden of Disease study, ambient PM pollution caused over

100,000 premature deaths in Russia in 2013 (GBD, 2016)."

GBD: The global burden of disease study 2013, Institute for Health Metrics and Evaluation, Seattle, WA. Available at http://ghdx.healthdata.org/gbd-data-tool (Last access: 12 July), 2016.

**Comment 2.** On p. 4, the authors discuss the movement to low sulfur fuel, but omit an important piece of information – low sulfur fuel is necessary for operation of the most effective emission control devices on diesel vehicles. This would be additional context that would be helpful to add.

Thank you. We updated the text as follows (L 121--124):

"Sulfur content of diesel fuel is an important factor in emission reductions. Diesel with high sulfur content (measured in parts per million or ppm) can destroy emission control devices, such as particulate filters. Availability of low sulfur diesel is an important prerequisite for the introduction of more stringent vehicle emission standards."

**Comment 3.** If the information is available, it would be informative to include the share of industry that is made up of small businesses rather than the number of people employed to give a better indication of the impact of the lack of reporting. (p. 5, lines 149).

Thank you for pointing this out.

Small businesses are not required to submit this information, yet they employ 11 million people (Fedstat, 2015f, a, b, e) **and produced more than 20% of goods and services (GKS, 2015a).**

GKS: Russian Statistical Yearbook 2015. Federal Statistics Service of the Russian Federation. Available at http://www.gks.ru/free_doc/doc_2015/year/year15.rar (last access: 26 February 2016), 2015a.

**Comment 4.** Authors should consider defining "fuel balance approach" at first mention rather than later in discussion in case readers are not familiar with this methodology. (p. 5, line 155)

We added to the text (Lines: 163-164):

However, their assessments both have their limitations because they do not use a fuel balance approach; *in other words they do not match diesel consumption by on-road vehicles and off-road engines with the production of diesel fuel in the country.*

**Comment 5.** It is not clear why there is discussion of gasoline vehicles in the section of distribution by emission standards. The authors note that they produce almost no BC, so could easily be left out or addressed with a sentence explaining that they produce almost no BC. If there is a reason the comparison of emission standards is important to the discussion, this should be clarified. (p. 6, lines 198-208)

Thank you for this suggestion. We deleted the discussion about gasoline vehicles. The updated text (L 224-226):

"Figure 2 shows the distribution of diesel vehicles by emission standard. Because gasoline

vehicles emit practically no BC, gasoline vehicles were differentiated from diesel vehicles in the

on-road fleet and not analyzed in this study."

**Comment 6.** For logical flow, authors should consider moving the active vehicles section (p. 7, lines 210-221) to immediately following the registered fleet section.

Thank you, we moved the *Active Vehicle* section after the *Registered Fleet* section. (For easy reading, the changes were not tracked).

**Comment 7.** The assumptions for speed and type of road traveled would benefit from further explanation (i.e., are these based on standard speeds/distribution of roads in Russia?) (p. 9, line 285)

The assumption on the speed in cities is based on actual speed in Moscow and other large cities. The assumptions on the average speed on rural roads and highways are based on maximum allowable (standard) speed on these types of roads in Russia.

The updated text (L 295-299):

"The assumption on the speed in cities is based on actual speed in Moscow and other large cities. The assumptions on the average speed on rural roads and highways are based on maximum allowable (standard) speed on these types of roads in Russia. The share of vehicle-kilometers traveled (vkt) on urban roads is taken from the ICCT Roadmap model (ICCT, 2015)..

The share of vkt on urban roads is 75% for cars, light commercial vehicles, and buses and 50% for trucks. The rest of VKT is divided by 40:60 between rural roads and highways."

**Comment 8**. The assumptions for controls on agricultural vehicles would benefit from further explanation (i.e., who no Stage 1?) (p. 11, line 351)

We assume that 95% of agricultural fleet has no emission controls and 5% meets Stage 2 standards. Only imported used tractors may have emission controls. In the European Union, Stage 1 emission standard was implemented in 1999 and Stage 2 implemented from 2001 to 2004. Stage 1 tractors might be too old for importing them to Russia so we assume that all imported tractors with emission controls meet Stage 2 standard. Stage 3 standards were phased in from 2006 to 2013, and these tractors are too new to be sold as used in 2014.

We updated the text as follow (L 363-364):

"Tractors imported from Western countries were assumed to have emission controls;

however, their share in the total agricultural fleet is very small (no more than 5 %).

**Comment 9.** For logical flow, recommend moving the paragraph on uncertainty regarding BC/PM ratios after the paragraph on activity data (p. 14). It appears that the BC/PM ratios are NOT a major source of uncertainty, so it would make more sense for the reader if this is discussed after the two major sources of uncertainty that are identified.

Thank you for the suggestion. We moved the paragraph (L 495-498).

Thank you for the useful comments and suggestions!

**Authors' comments on Review #2**

Atmos. Chem. Phys. Discuss.,
doi:10.5194/acp-2016-475-RC2, 2016

**"Russia's black carbon emissions: focus on diesel sources" by N. Kholod et al.**

We thank the referee for the helpful comments.

**Comment 1:** Lines 181-184: authors claim "Russia does not have large-scale production of diesel passenger cars", but also say 98% of diesel cars were either imported or produced in Russia by foreign companies. Not sure how this affects the overall emissions, but what fraction of on-road diesel cars are made in Russia by Russian or non-Russian companies? (Also, since this paper focuses on diesel cars, the line about "foreign-make cars, both gasoline and diesel" is superfluous.)

**Answer:** The Russian vehicle registry shows the following data about registered cars: vehicle type, manufacturer, year of manufacturing, fuel type, emission (Euro) class, and ownership type. The registry data do not allow for distinguishing between imported vehicles and those produced in Russia by foreign companies.

Changes to the text:
We have deleted the phrase "Russia does not have large-scale production of diesel passenger cars" to avoid confusion.

Thank you for pointing out that the line about "foreign-make cars, both gasoline and diesel" is superfluous. We have deleted the line.

**Comment 2:** Lines 203-204: what higher emissions standards do imported diesel vehicles meet? Euro 6? Or were imported vehicles always produced to meet a higher standard than necessary for Russia?

**Answer:** Russian mainly imports diesel vehicles from Japan, the European Union, and the United States, with a smaller number of vehicles from South Korea and China. The imported vehicles might meet the emissions standards of the manufacturing countries. Russia adopted European emissions standards about 10 years after the EU, Japan, and the US. As a result, vehicles produced abroad and imported to Russia might meet higher emission standards. However, there is a chance that vehicles produced in foreign countries for the Russian market can only meet the Russian standards. Imported USED vehicles do meet higher emissions standards than necessary in Russia. For example, the EU implemented the Euro 5 standard in 2005 and Euro 6 in 2014 while Russia implemented the Euro 5 standard only in 2016. Among all imported used cars to Russia in 2015, over 70% were vehicles made by Toyota, Nissan, Volkswagen, and Renault.

As a result, when analyzing the distribution by emission standard we rely on the data from the registry.

**Correction to the text:**
We have deleted this paragraph per comment from the first reviewer.

**Comment 3:** Lines 240-251: The authors rely on the Bond et al. (2004) assumption of super-emitter fraction as 10%, even though they cite several more recent studies that show superemitters can be as high as 13-15% of the fleet, even in California. Given the lack of studies in Russia, and the authors' literature survey of the Russian fleet (36% of trucks and 23% of buses older than 20 years), using the old Bond et al. (2004) assumption will likely bias their emissions inventory low as the authors acknowledge at the end. The authors should investigate the sensitivity of their results to this fraction, and perhaps try higher values (15-30%) for the super-emitter fraction.

Answer:
We have modified the text to assume that the share of superemitters is 15%. In the sensitivity analysis, we assume that the share of superemitters in the diesel fleet ranges between 10% and 20%.

Changes to the text: please see answer to Comment 7.

**Comment 4:** Lines 286-287: What is the basis for their assumption of 40-20-40 on urban roads, rural roads, and highways?

Answer:
Thank you for pointing this out. The first reviewer also asked the same question. We updated our assumptions on the distribution of vehicle-kilometers traveled on urban roads, rural roads, and highways. The share of vehicle-kilometers traveled on urban roads is taken from the ICCT Roadmap model (http://www.theicct.org/global-transportation-roadmap-model), and the rest is divided by 40:60 between rural roads and highways (our assumption based on expert judgement).

Changes to the text (L295-299):
"The share of vehicle-kilometers traveled (vkt) on urban roads is taken from the ICCT Roadmap model. The share of vkt on urban roads is 75% for cars, light commercial vehicles, and buses and 50% for trucks. The rest of VKT the rest is divided by 40:60 between rural roads and highways."

**Comment 5:** The authors use NIIAT data for on-road emission factors, but the actual source of that data is not clear – are these based on measurements or on estimates based on emissions

standards? The authors present the data used; a brief explanation of the source methodology will be helpful, since these NIIAT publications do not appear to be easily accessible online.

Answer:
The NIIAT data on emission factors for Russian models are based on measurement. NIIAT has been working on emission methodologies since the 1980s, and tests for vehicles without emission controls (Euro 0), Euro 1 , and Euro 2 were conducted together with the Environmental Department of the Scientific and Research Vehicle Testing Center located in the Moscow region (Donchenko, V., Kunin, Y., Ruzski, A., Vizhenski, V., 2014. Evaluation of road transport effect on atmospheric air: method of emission computations and use of results. Transport Research Arena, Paris. Available at
http://tra2014.traconference.eu/papers/pdfs/TRA2014_Fpaper_19875.pdf).
For foreign models, NIIAT uses emission factors from the European EMEP/CORINAIR guidebook. For its emission calculation methodologies, NIIAT blended emission factors for Russian and European vehicles to reflect the composition of the Russian on-road fleet.

The NIIAT methodologies are not available in English. We worked directly with NIIAT experts and received methodological explanations during multiple meetings. We also presented the results of our emission calculations at a meeting in the NIIAT office in Moscow.

Changes to the text (L 89-92):
"Based on vehicle driving tests conducted with Scientific and Research Vehicle Testing Center, NIIAT has developed emission factors for Russian models. For foreign-made vehicles, NIIAT relies on data from the European EMEP/CORINAIR guidebook. Thus, Russian-specific emission factors for $PM_{2.5}$ in the NIIAT methodologies are based on the average for every vehicle type and emission class on Russian roads."

**Comment 6:** The conclusions should note that the results exclude military diesel usage emissions; in particular, these could be large sources of sulfate PM, and possibly also BC.

Answer: we updated the text in conclusion as follows (L 542-543):
"These results do not include emissions from military diesel usage. Military vehicles can be a large source of BC emissions given that they use high-sulfur diesel."

**Comment 7:** While the authors present a comprehensive list of potential uncertainties with their emissions inventory estimate, they don't propagate the uncertainties through, which would be helpful. From their list, it appears the emission factors could produce uncertainties of +/-30% or so, while the bias due to low super-emitter fraction (10% when 15-20% might be more appropriate) could increase the overall BC estimate by as much as 40%!

Answer: For sensitivity analysis we assume that the share of superemitters in the total diesel fleet is in the range of 10%-20% with the central estimate of 15%. We also propagated the uncertainties for on-road vehicles and off-road diesel sources.

Changes to the text (514-524):
"For on-road vehicles, three major sources of uncertainty were considered: the share of superemitters in the fleet, average annual distance traveled, and emissions factors for normal vehicles and superemitters. Supplement Table S10 shows the assumption for uncertainty calculations for on-road vehicles.

The central value of BC emissions from on-road vehicles in 2014 is 20.7 Gg with an uncertainty range of -10.2 Gg and + 7.3 Gg. The central value of OC emissions is 10.5 Gg with an uncertainty range of -4.2 Gg and + 3.2 Gg.

Supplement Table S10. Uncertainty estimates for BC and OC emissions from on-road vehicles

|  | Central | Minimum | Maximum |
|---|---|---|---|
| Share of superemitters | 15% | 10% | 20% |
| Annual distance traveled, km | Avtostat | NIIAT | Avtostat |
|   Cars | 15 000 | 15 000 | 15 000 |
|   LCVs | 55 000 | 30 000 | 55 000 |
|   Trucks | 63 000 | 45 000 | 63 000 |
|   Buses | 65 000 | 50 000 | 65 000 |
| PM emissions factor | COPERT | COPERT -20% | COPERT +20% |
| BC/PM speciation ratio | COPERT | COPERT -10% | COPERT +10% |
| Emissions, Gg |  |  |  |
| BC normal | 11.8 | 7.1 | 12.3 |
| BC superemitters | 8.9 | 3.4 | 15.7 |
| **BC total** | **20.7** | **10.5** | **28.0** |
| OC normal | 5.6 | 4.3 | 5.0 |
| OC superemitters | 4.9 | 2.1 | 8.7 |
| **OC total** | **10.5** | **6.4** | **13.7** |

The uncertainty in BC emissions from off-road sources is estimated in the range from 19.2 Gg to 42.1 Gg (or -33%/+48%) with the central value of 28.5 Gg. OC emissions from off-road engines are in the range from 4.5 Gg to 9.8 Gg with the central value of 6.7 Gg.

The total emissions from diesel sources in Russia are estimated to be 49.2 Gg of BC and 17.2 Gg of OC in 2014."

**Comment 8:** One final concern is that the current submission has no explanation of differences between this paper, and the on-road BC emissions estimate published earlier by the first author

(Kholod and Evans, http://dx.doi.org/10.1016/j.envsci.2015.10.017 ) While the current paper is more detailed, the bottom line figure appears the same - in 2015, Figure 1 of Kholod and Evans shows 20,000 tons of BC from on-road Russian sources similar to the current paper. Maybe the complicated model of the current submission is not needed?!

Answer:
There several important differences between this article and Kholod and Evans (2016).

- Kholod and Evans (2016) use the Global Change Assessment Model (GCAM) to build a forecast for BC emissions from on-road transport (Figure 1). The model calculates emissions in 5-year time intervals. Though the model is a powerful tool to project BC emissions, the model does not distinguish between diesel and gasoline vehicles and does not show the BC distribution by emission standards.
- The current study uses activity-based emissions factors (g/kg fuel). As we show in the article, large uncertainty exists in the fuel consumption by on-road vehicles (in the range from 11 million tons to 22 million tons). In the current study, the emission calculations for on-road vehicles do not use fuel data. Instead we use data on annual distance traveled and activity-based emission factors (g/km). This approach allows us to calculate emissions by vehicle type and emission standard. We also account for superemitters.

Changes to the text (L470-473):

[revised manuscript text omitted]

* By convention, light-duty vehicles are marked with Arabic numerals while Roman numbers are used for heavy-duty vehicles (trucks and buses).

**Table S2**. COPERT and NIIAT emission factors and BC speciation for hot operation stage.

| Type | Subcategory | COPERT EF, g/km | | | Blended NIIAT EFs, g/km | | | EC/PM* | OC/EC* |
|---|---|---|---|---|---|---|---|---|---|
| **Cars** | | Urban | Rural | Highway | Urban | Rural | Highway | | |
| Euro 0 | | 0.271 | 0.199 | 0.146 | 0.250 | 0.150 | 0.170 | 0.55 | 0.70 |
| Euro 1 | | 0.07 | 0.057 | 0.087 | 0.073 | 0.040 | 0.050 | 0.70 | 0.40 |
| Euro 2 | | 0.058 | 0.047 | 0.045 | 0.073 | 0.040 | 0.050 | 0.80 | 0.23 |
| Euro 3 | | 0.035 | 0.029 | 0.038 | 0.053 | 0.030 | 0.030 | 0.85 | 0.15 |
| Euro 4 | | 0.034 | 0.029 | 0.025 | 0.016 | 0.090 | 0.090 | 0.87 | 0.13 |
| Euro 5 | | 0.003 | 0.002 | 0.002 | 0.004 | 0.002 | 0.002 | 0.20 | 2.00 |
| **LCV** | | | | | | | | | |
| Euro 0 | | 0.281 | 0.285 | 0.337 | 0.290 | 0.210 | 0.230 | 0.55 | 0.70 |
| Euro 1 | | 0.099 | 0.07 | 0.118 | 0.087 | 0.060 | 0.100 | 0.70 | 0.40 |
| Euro 2 | | 0.099 | 0.07 | 0.118 | 0.087 | 0.060 | 0.100 | 0.80 | 0.23 |
| Euro 3 | | 0.066 | 0.047 | 0.079 | 0.057 | 0.040 | 0.060 | 0.85 | 0.15 |
| Euro 4 | | 0.035 | 0.024 | 0.041 | 0.033 | 0.020 | 0.030 | 0.87 | 0.13 |
| Euro 5 | | 0.002 | 0.001 | 0.001 | 0.002 | 0.001 | 0.002 | 0.20 | 2.00 |
| **Trucks** | | | | | | | | | |
| Euro 0 | <=7,5 t | 0.4 | 0.297 | 0.211 | 0.543 | 0.180 | 0.180 | 0.50 | 0.80 |
| Euro I | <=7,5 t | 0.157 | 0.116 | 0.09 | 0.360 | 0.140 | 0.140 | 0.65 | 0.40 |
| Euro II | <=7,5 t | 0.069 | 0.056 | 0.064 | 0.220 | 0.080 | 0.080 | 0.65 | 0.40 |
| Euro III | <=7,5 t | 0.082 | 0.061 | 0.04 | 0.153 | 0.060 | 0.060 | 0.70 | 0.30 |
| Euro IV | <=7,5 t | 0.017 | 0.014 | 0.015 | 0.030 | 0.010 | 0.010 | 0.75 | 0.25 |
| Euro V | <=7,5 t | 0.021 | 0.018 | 0.015 | 0.030 | 0.010 | 0.010 | 0.75 | 0.25 |
| Euro 0 | 7,5 - 12 t | 0.423 | 0.301 | 0.201 | 0.893 | 0.400 | 0.400 | 0.50 | 0.80 |
| Euro I | 7,5 - 12 t | 0.262 | 0.182 | 0.129 | 0.640 | 0.330 | 0.330 | 0.65 | 0.40 |
| Euro II | 7,5 - 12 t | 0.115 | 0.087 | 0.098 | 0.230 | 0.100 | 0.100 | 0.65 | 0.40 |
| Euro III | 7,5 - 12 t | 0.133 | 0.095 | 0.062 | 0.153 | 0.060 | 0.060 | 0.70 | 0.30 |
| Euro IV | 7,5 - 12 t | 0.027 | 0.021 | 0.02 | 0.030 | 0.010 | 0.010 | 0.75 | 0.25 |
| Euro V | 7,5 - 12 t | 0.034 | 0.026 | 0.021 | 0.030 | 0.010 | 0.010 | 0.75 | 0.25 |

| | | | | | | | | |
|---|---|---|---|---|---|---|---|---|
| Euro 0 | 12 - 14 t | 0.452 | 0.32 | 0.232 | 1.073 | 0.550 | 0.550 | 0.50 | 0.80 |
| Euro I | 12 - 14 t | 0.28 | 0.199 | 0.147 | 0.697 | 0.480 | 0.480 | 0.65 | 0.40 |
| Euro II | 12 - 14 t | 0.128 | 0.095 | 0.109 | 0.310 | 0.180 | 0.180 | 0.65 | 0.40 |
| Euro III | 12 - 14 t | 0.141 | 0.099 | 0.071 | 0.193 | 0.130 | 0.130 | 0.70 | 0.30 |
| Euro IV | 12 - 14 t | 0.03 | 0.023 | 0.02 | 0.040 | 0.020 | 0.020 | 0.75 | 0.25 |
| Euro V | 12 - 14 t | 0.036 | 0.028 | 0.023 | 0.040 | 0.020 | 0.020 | 0.75 | 0.25 |
| Euro 0 | > 14 t | 0.625 | 0.439 | 0.29 | 1.073 | 0.550 | 0.550 | 0.50 | 0.80 |
| Euro I | > 14 t | 0.386 | 0.271 | 0.175 | 0.697 | 0.480 | 0.480 | 0.65 | 0.40 |
| Euro II | > 14 t | 0.164 | 0.118 | 0.129 | 0.310 | 0.180 | 0.180 | 0.65 | 0.40 |
| Euro III | > 14 t | 0.199 | 0.139 | 0.087 | 0.193 | 0.130 | 0.130 | 0.70 | 0.30 |
| Euro IV | > 14 t | 0.039 | 0.029 | 0.023 | 0.040 | 0.020 | 0.020 | 0.75 | 0.25 |
| Euro V | > 14 t | 0.049 | 0.037 | 0.028 | 0.040 | 0.020 | 0.020 | 0.75 | 0.25 |
| **Buses** | | | | | | | | | |
| Euro 0 | <=15 t | 0.858 | 0.574 | 0.388 | 0.880 | 0.270 | 0.295 | 0.50 | 0.80 |
| Euro I | <=15 t | 0.294 | 0.221 | 0.173 | 0.650 | 0.215 | 0.230 | 0.65 | 0.40 |
| Euro II | <=15 t | 0.142 | 0.114 | 0.107 | 0.398 | 0.195 | 0.175 | 0.65 | 0.40 |
| Euro III | <=15 t | 0.146 | 0.11 | 0.099 | 0.197 | 0.105 | 0.100 | 0.70 | 0.30 |
| Euro IV | <=15 t | 0.035 | 0.027 | 0.022 | 0.040 | 0.025 | 0.025 | 0.75 | 0.25 |
| Euro V | <=15 t | 0.042 | 0.031 | 0.038 | 0.037 | 0.025 | 0.025 | 0.75 | 0.25 |
| Euro 0 | 15 - 18 t | 0.767 | 0.52 | 0.312 | 1.523 | 0.430 | 0.500 | 0.50 | 0.80 |
| Euro I | 15 - 18 t | 0.412 | 0.294 | 0.217 | 0.890 | 0.310 | 0.400 | 0.65 | 0.40 |
| Euro II | 15 - 18 t | 0.197 | 0.157 | 0.138 | 0.680 | 0.310 | 0.270 | 0.65 | 0.40 |
| Euro III | 15 - 18 t | 0.195 | 0.148 | 0.108 | 0.250 | 0.130 | 0.120 | 0.70 | 0.30 |
| Euro IV | 15 - 18 t | 0.049 | 0.037 | 0.027 | 0.050 | 0.030 | 0.030 | 0.75 | 0.25 |
| Euro V | 15 - 18 t | 0.055 | 0.042 | 0.036 | 0.050 | 0.030 | 0.030 | 0.75 | 0.25 |
| Euro 0 | >18 t | 0.957 | 0.675 | 0.395 | 1.523 | 0.430 | 0.500 | 0.50 | 0.80 |
| Euro I | >18 t | 0.517 | 0.37 | 0.227 | 0.757 | 0.310 | 0.400 | 0.65 | 0.40 |
| Euro II | >18 t | 0.265 | 0.212 | 0.174 | 0.583 | 0.310 | 0.270 | 0.65 | 0.40 |
| Euro III | >18 t | 0.24 | 0.17 | 0.125 | 0.250 | 0.130 | 0.120 | 0.70 | 0.30 |
| Euro IV | >18 t | 0.06 | 0.045 | 0.029 | 0.050 | 0.030 | 0.030 | 0.75 | 0.25 |
| Euro V | >18 t | 0.066 | 0.049 | 0.04 | 0.050 | 0.030 | 0.030 | 0.75 | 0.25 |

Sources: (Emisia, 2015;NIIAT, 2012).

\* EC/PM and OC/EC speciation factors are derived from the COPERT model.

**Table S3**. Number of active diesel vehicles by type and emission standard in Russia, 2014.

|          | Cars       | LCVs    | Trucks    | Buses   |
|----------|------------|---------|-----------|---------|
| Euro 0   | 100 620    | 150 345 | 470 737   | 24 994  |
| Euro 1   | 103 529    | 119 732 | 43 093    | 10 038  |
| Euro 2   | 113 576    | 94 135  | 173 337   | 17 541  |
| Euro 3   | 144 298    | 140 923 | 293 520   | 49 725  |
| Euro 4   | 691 199    | 353 189 | 271 189   | 19 481  |
| Euro 5   | 329 941*   | 58 703  | 77 404    | 1 615   |
| Total    | 1 483 163  | 917 027 | 1 329 280 | 123 394 |

 * - includes 2110 Euro 6 cars
Calculated based on (Avtostat, 2015).

**Table S4.** The annual average distance traveled by type of vehicles, thousand kmyr$^{-1}$.

|              | Cars                                                                             | LCVs | Trucks                                        | Buses                                                                                                                         |
|--------------|----------------------------------------------------------------------------------|------|-----------------------------------------------|------------------------------------------------------------------------------------------------------------------------------|
| NIIAT (1998) | 15
10 for 5-year old Russian cars
10 for 10-year old foreign cars          |      | 35 in cities
60 suburban
100 intercity  | Russian:
50 in cities
65 suburban
80 intercity
Foreign:
60 in cities
80 suburban
105 intercity          |
| NIIAT (2008) | 14-16 owned by individuals
25-30 owned by companies                           |      | 30-40                                         | 40-50                                                                                                                        |
| Avtostat (2010) | 16.7
15.3 Russian
18 foreign-made                                       | 55   | 63                                            | 65                                                                                                                           |
| ICCT (2015)  | 10                                                                               | 10   | 13-38                                         | 56                                                                                                                           |

**Table S5.** Average number kilometers traveled by type of vehicles.

| Vehicle type    | Subsector       | Emission standard              | Annual kilometers traveled |
|-----------------|-----------------|--------------------------------|----------------------------|
|                 | Diesel 1,4 - 2,0 l | Conventional                | 10 000                     |
|                 | Diesel 1,4 - 2,0 l | PC Euro 1 - 91/441/EEC      | 10 000                     |
| Passenger Cars  | Diesel 1,4 - 2,0 l | PC Euro 2 - 94/12/EEC       | 15 000                     |
|                 | Diesel 1,4 - 2,0 l | PC Euro 3 - 98/69/EC Stage2000 | 15 000                  |
|                 | Diesel 1,4 - 2,0 l | PC Euro 4 - 98/69/EC Stage2005 | 18 000                  |

| | Diesel 1,4 - 2,0 l | PC Euro 5 - EC 715/2007 | 20 000 |
|---|---|---|---|
| Light Commercial Vehicles | Diesel <3,5 t | Conventional | 37 000 |
| | Diesel <3,5 t | LD Euro 1 - 93/59/EEC | 37 000 |
| | Diesel <3,5 t | LD Euro 2 - 96/69/EEC | 55 000 |
| | Diesel <3,5 t | LD Euro 3 - 98/69/EC Stage2000 | 55 000 |
| | Diesel <3,5 t | LD Euro 4 - 98/69/EC Stage2005 | 66 000 |
| | Diesel <3,5 t | LD Euro 5 - 2008 Standards | 73 000 |
| Heavy Duty Trucks | Rigid <=7,5 t | Conventional | 42 000 |
| | Rigid <=7,5 t | HD Euro I - 91/542/EEC Stage I | 42 000 |
| | Rigid <=7,5 t | HD Euro II - 91/542/EEC Stage II | 63 000 |
| | Rigid <=7,5 t | HD Euro III - 2000 Standards | 63 000 |
| | Rigid <=7,5 t | HD Euro IV - 2005 Standards | 75 000 |
| | Rigid <=7,5 t | HD Euro V - 2008 Standards | 84 000 |
| | Rigid 7,5 - 12 t | Conventional | 42 000 |
| | Rigid 7,5 - 12 t | HD Euro I - 91/542/EEC Stage I | 42 000 |
| | Rigid 7,5 - 12 t | HD Euro II - 91/542/EEC Stage II | 63 000 |
| | Rigid 7,5 - 12 t | HD Euro III - 2000 Standards | 63 000 |
| | Rigid 7,5 - 12 t | HD Euro IV - 2005 Standards | 75 000 |
| | Rigid 7,5 - 12 t | HD Euro V - 2008 Standards | 84 000 |
| | Rigid 12 - 14 t | Conventional | 42 000 |
| | Rigid 12 - 14 t | HD Euro I - 91/542/EEC Stage I | 42 000 |
| | Rigid 12 - 14 t | HD Euro II - 91/542/EEC Stage II | 63 000 |
| | Rigid 12 - 14 t | HD Euro III - 2000 Standards | 63 000 |
| | Rigid 12 - 14 t | HD Euro IV - 2005 Standards | 75 000 |
| | Rigid 12 - 14 t | HD Euro V - 2008 Standards | 84 000 |
| | Rigid 14 - 20 t | Conventional | 42 000 |
| | Rigid 14 - 20 t | HD Euro I - 91/542/EEC Stage I | 42 000 |
| | Rigid 14 - 20 t | HD Euro II - 91/542/EEC Stage II | 63 000 |
| | Rigid 14 - 20 t | HD Euro III - 2000 Standards | 63 000 |
| | Rigid 14 - 20 t | HD Euro IV - 2005 Standards | 75 000 |
| | Rigid 14 - 20 t | HD Euro V - 2008 Standards | 84 000 |
| Buses | Urban Buses Midi <=15 t | Conventional | 43 000 |
| | Urban Buses Midi <=15 t | HD Euro I - 91/542/EEC Stage I | 43 000 |
| | Urban Buses Midi <=15 t | HD Euro II - 91/542/EEC Stage II | 65 000 |
| | Urban Buses Midi <=15 t | HD Euro III - 2000 Standards | 65 000 |
| | Urban Buses Midi <=15 t | HD Euro IV - 2005 Standards | 78 000 |
| | Urban Buses Midi <=15 t | HD Euro V - 2008 Standards | 87 000 |
| | Urban Buses Standard 15 - 18 t | Conventional | 43 000 |
| | Urban Buses Standard 15 - 18 t | HD Euro I - 91/542/EEC Stage I | 43 000 |
| | Urban Buses Standard 15 - 18 t | HD Euro II - 91/542/EEC Stage II | 65 000 |
| | Urban Buses Standard 15 - 18 t | HD Euro III - 2000 Standards | 65 000 |
| | Urban Buses Standard 15 - 18 t | HD Euro IV - 2005 Standards | 78 000 |

| | | |
|---|---|---|
| Urban Buses Standard 15 - 18 t | HD Euro V - 2008 Standards | 87 000 |
| Urban Buses Articulated >18 t | Conventional | 43 000 |
| Urban Buses Articulated >18 t | HD Euro I - 91/542/EEC Stage I | 43 000 |
| Urban Buses Articulated >18 t | HD Euro II - 91/542/EEC Stage II | 65 000 |
| Urban Buses Articulated >18 t | HD Euro III - 2000 Standards | 65 000 |
| Urban Buses Articulated >18 t | HD Euro IV - 2005 Standards | 78 000 |
| Urban Buses Articulated >18 t | HD Euro V - 2008 Standards | 87 000 |

**Results of emission calculations**

The COPERT 4 model with NIIAT emission factors. The assumptions on average annual kilometers traveled remain the same.

**Table S6.** OC emissions from the adjusted diesel fleet with superemitters (Gg).

|        | Cars | LCVs | Trucks | Buses | Total |
|--------|------|------|--------|-------|-------|
| Euro 0 | 0.07 | 0.47 | 2.83   | 0.20  | 3.58  |
| Euro 1 | 0.02 | 0.11 | 0.14   | 0.05  | 0.31  |
| Euro 2 | 0.03 | 0.10 | 0.49   | 0.10  | 0.72  |
| Euro 3 | 0.02 | 0.08 | 0.49   | 0.12  | 0.70  |
| Euro 4 | 0.03 | 0.13 | 0.09   | 0.01  | 0.26  |
| Euro 5 | 0.01 | 0.01 | 0.03   | 0.00  | 0.05  |
| Total  | 0.18 | 0.88 | 4.07   | 0.48  | 5.62  |

**Table S7**. BC emissions, assuming that all registered trucks and buses use diesel fuel (Gg).

|        | Trucks | Buses | Total |
|--------|--------|-------|-------|
| Euro 0 | 33.2   | 3.8   | 80.2  |
| Euro 1 | 1.1    | 0.3   | 3.6   |
| Euro 2 | 2.5    | 0.8   | 13.6  |
| Euro 3 | 2.3    | 0.8   | 11.4  |
| Euro 4 | 0.4    | 0.1   | 5.8   |
| Euro 5 | 0.1    | 0.0   | 0.2   |
| Total  | 39.7   | 5.7   | 114.8 |

**Table S8.** BC emissions from all registered diesel vehicles (Gg).

|        | Cars | LCVs | Trucks | Buses | Total |
|--------|------|------|--------|-------|-------|
| Euro 0 | 0.36 | 1.85 | 15.39  | 0.66  | 18.26 |
| Euro 1 | 0.10 | 0.54 | 1.06   | 0.20  | 1.91  |
| Euro 2 | 0.18 | 0.68 | 2.33   | 0.32  | 3.51  |
| Euro 3 | 0.16 | 0.66 | 2.27   | 0.42  | 3.52  |
| Euro 4 | 0.27 | 1.03 | 0.42   | 0.04  | 1.77  |
| Euro 5 | 0.01 | 0.00 | 0.12   | 0.00  | 0.13  |
| Total  | 1.08 | 4.77 | 21.58  | 1.66  | 29.09 |

**Table S9.** BC emissions from the adjusted diesel fleet without accounting for superemitters (Gg).

|        | Cars | LCVs | Trucks | Buses | Total |
|--------|------|------|--------|-------|-------|
| Euro 0 | 0.20 | 1.30 | 6.16   | 0.46  | 8.11  |
| Euro 1 | 0.08 | 0.41 | 0.53   | 0.16  | 1.18  |
| Euro 2 | 0.14 | 0.54 | 1.40   | 0.29  | 2.37  |
| Euro 3 | 0.14 | 0.56 | 1.71   | 0.42  | 2.82  |
| Euro 4 | 0.24 | 0.98 | 0.38   | 0.05  | 1.65  |
| Euro 5 | 0.01 | 0.00 | 0.12   | 0.00  | 0.13  |
| Total  | 0.80 | 3.79 | 10.28  | 1.39  | 16.26 |

**Table S10.** Uncertainty estimates for BC and OC emissions from on-road vehicles.

|                              | Central  | Minimum      | Maximum      |
|------------------------------|----------|--------------|--------------|
| Share of superemitters       | 15%      | 10%          | 20%          |
| Annual distance traveled, km | Avtostat | NIIAT        | Avtostat     |
| Cars                         | 15 000   | 15 000       | 15 000       |
| LCVs                         | 55 000   | 30 000       | 55 000       |
| Trucks                       | 63 000   | 45 000       | 63 000       |
| Buses                        | 65 000   | 50 000       | 65 000       |
| PM emissions factor          | COPERT   | COPERT -20%  | COPERT +20%  |
| BC/PM speciation ratio       | COPERT   | COPERT -10%  | COPERT +10%  |
| Emissions, Gg                |          |              |              |
| BC normal                    | 11.8     | 7.1          | 12.3         |
| BC superemitters             | 8.9      | 3.4          | 15.7         |
| **BC total**                 | **20.7** | **10.5**     | **28.0**     |
| OC normal                    | 5.6      | 4.3          | 5.0          |
| OC superemitters             | 4.9      | 2.1          | 8.7          |
| **OC total**                 | **10.5** | **6.4**      | **13.7**     |